**Data Availability Statement:** Permission to access, handle and analyze the data of the present

# Healthcare resource utilization in patients with treatment-resistant depression—A Danish national registry study

Kristoffer Jarlov Jensen[1]*, Frederikke Hørdam Gronemann[2], Mikkel Zöllner Ankarfeldt[1], Espen Jimenez-Solem[1,3,4], Sarah Alulis[5], Jesper Riise[5], Nikolaj Bødker[5], Merete Osler[2,6], Janne Petersen[1,7]

1 Copenhagen Phase IV Unit (Phase4CPH), Department of Clinical Pharmacology and Center for Clinical Research and Prevention, Copenhagen University Hospital Bispebjerg and Frederiksberg, Copenhagen, Denmark, 2 Section for Population-based Epidemiology, Center for Clinical Research and Prevention, Frederiksberg Hospital, Frederiksberg, Denmark, 3 Department of Clinical Pharmacology, Bispebjerg Hospital, Copenhagen, Denmark, 4 Faculty of Health and Medical Sciences, University of Copenhagen, Copenhagen, Denmark, 5 Janssen Cilag A/S, Birkerød, Denmark, 6 Section for Epidemiology, Department of Public Health, University of Copenhagen, Copenhagen, Denmark, 7 Section for Biostatistics, Department of Public Health, University of Copenhagen, Copenhagen, Denmark

* Kristoffer.Jarlov.Jensen@regionh.dk

## Abstract

### Objectives

To investigate healthcare resource utilization (HRU) and associated costs by depression severity and year of diagnosis among patients with treatment-resistant depression (TRD) in Denmark.

### Methods

Including all adult patients with a first-time hospital contact for major depressive disorder (MDD) in 1996–2015, TRD patients were defined at the second shift in depression treatment (antidepressant medicine or electroconvulsive therapy) and matched 1:2 with non-TRD patients. The risk of utilization and amount of HRU and associated costs including medicine expenses 12 months after the TRD-defining date were reported, comparing TRD patients with non-TRD MDD patients.

### Results

Identifying 25,321 TRD-patients matched with 50,638 non-TRD patients, the risk of psychiatric hospitalization following TRD diagnosis was 138.4% (95%-confidence interval: 128.3–149.0) higher for TRD patients than for non-TRD MDD patients. The number of hospital bed days and emergency department (ED) visits were also higher among TRD patients, with no significant difference for somatic HRU. Among patients who incurred healthcare costs, the associated HRU costs for TRD patients were 101.9% (97.5–106.4) higher overall, and 55.2% (50.9–59.6) higher for psychiatric services than those of non-TRD patients. The relative differences in costs for TRD-patients vs non-TRD patients were greater for patients with

study was obtained from the Danish Data Protection Agency (www.datatilsynet.dk) via the local representing unit in the Capital Region (protocol number CSU-FCFS-2016-012, I-Suite #: 04818). Data access to the specific registries was furthermore applied for through Statistics Denmark and the Danish Health Data Authority, respectively. Data analyzed for the present study is in the domain of Statistics Denmark (www.dst.dk) and cannot be transferred to third party.

**Funding:** The study was financed by Janssen Cilag A/S. The funders contributed to conception of the study and writing of the manuscript.

**Competing interests:** The study was performed by the Copenhagen Phase IV Unit (Phase4CPH) and was financed by Janssen Cilag A/S, which holds license to antidepressant medicine. Via Phase4CPH, JP, EJS and MZA have performed other studies regarding antidepressants involving funding from Janssen Cilag and Eli Lilly, while KJJ, FHG and MO have performed other studies regarding antidepressants involving funding from Janssen Cilag. All funds were given to their institution. JR, NB and SA are employees of Janssen Cilag. Janssen Cilag A/S and its employees did not have access to the registry data located on the servers of Statistics Denmark and had therefore no influence on the data management and data analysis of the present work. The authors at Copenhagen Phase IV Unit retained editorial control of the manuscript. This does not alter our adherence to PLOS ONE policies on sharing data and materials.

mild depression and tended to increase over the study period (1996–2015), particularly for acute hospitalizations and ED visits.

## Limitations

TRD was defined by prescription patterns besides ECT treatments.

## Conclusion

TRD was associated with increased psychiatric-related HRU. Particularly the difference in acute hospitalizations and ED visits between TRD and non-TRD patients increased over the study period.

## Introduction

Major depressive disorder (MDD) is a leading cause of disability worldwide [1] with an estimated global prevalence of 4.4% [2], and a lifetime prevalence of 13% in Europe [3], placing a substantial economic burden on patients and society [4]. A significant proportion of patients with MDD do not respond adequately to anti-depressant treatment. Treatment-resistant depression (TRD) is often defined as a patient's failure to respond to at least two consecutive antidepressant treatments in a single depressive episode, taken at adequate doses for an adequate length of time [5], though no consensus of adequacy of dose and duration exists [6]. It is a complex condition associated with comorbidities [7], higher risk of hospitalization and longer admission [8], increased suicidality and self-harm [9–11], increased risk of premature workforce exit [12], reduced quality of life, and poorer prognosis [13, 14] compared with non-TRD MDD patients. A recent Danish nation-wide cohort study found that 15.8% of MDD patients developed TRD within 12 months of first hospital contact for major depression, corresponding to an incidence of 187.7 (185.7; 189.9) per 1,000 person-years [15].

Two reviews of healthcare resource utilization (HRU) in depression from 2014 and 2019 [16, 17] found that TRD patients experience higher costs than non-TRD patients with the majority of studies being United States (US)-based, only one study was from Scandinavia [18].

The tax-financed healthcare system in Denmark offering universal access to all residents in Denmark differs significantly from the US healthcare system [19]. Therefore, extrapolating HRU evidence and associated costs from the US to Denmark is of limited value.

The primary study aim was to assess the HRU and the associated healthcare costs, categorized by healthcare service, in TRD patients compared with non-TRD MDD patients using national registers in Denmark. The secondary aim was to study the potential HRU differences between TRD and non-TRD patients, stratified by depression severity and year of diagnosis.

## Methods

### MDD population

The MDD population included all patients living in Denmark aged 18 or older with a first-time in- or outpatient hospital contact with an MDD diagnosis between January 1, 1996 and December 31, 2015. Hospital contacts and diagnoses were identified in the Danish Psychiatric Central Research Register [20] and the Danish National Patient Register [21]. Diagnoses were identified using the International Classification of Diseases, Tenth Revision (ICD-10) codes F32.0–F32.9 and F33.0–F33.9. Patients with a prior diagnosis of comorbid bipolar affective disorder (ICD-10: F30.x, F31.x; ICD-8: 296.19, 296.39, 296.89, 296.99, 298.19), other affective

mood disorders (ICD-10: F38.x, F39.x), persistent mood disorder (ICD-10: F34.x), schizophrenia (ICD-10: F20.x, F21.x F22.x, F23.x, F24.x, F25.x, F28.x, F29.x; ICD-8: 295.x9, 297.x9, 298.29–298.99, 299.04, 299.05, 299.09), or dementia (ICD-10: DF00.x–F03.x, DG30.x, DR54.9; ICD-8: 290.x) were not included, as these diseases may require antidepressant use (i.e. the anti-depressants investigated herein have alternative indications) or may cause non-response to antidepressants. Patients with prevalent TRD (see TRD definition below) at the date of MDD diagnosis were also excluded (Fig 1).

## TRD population

TRD was defined at the second shift in antidepressant treatment, that occurred within one year prior to one year after the MDD diagnosis. In line with previous studies [15, 22], treatment shifts were considered as either a change in antidepressant drug (from one chemical

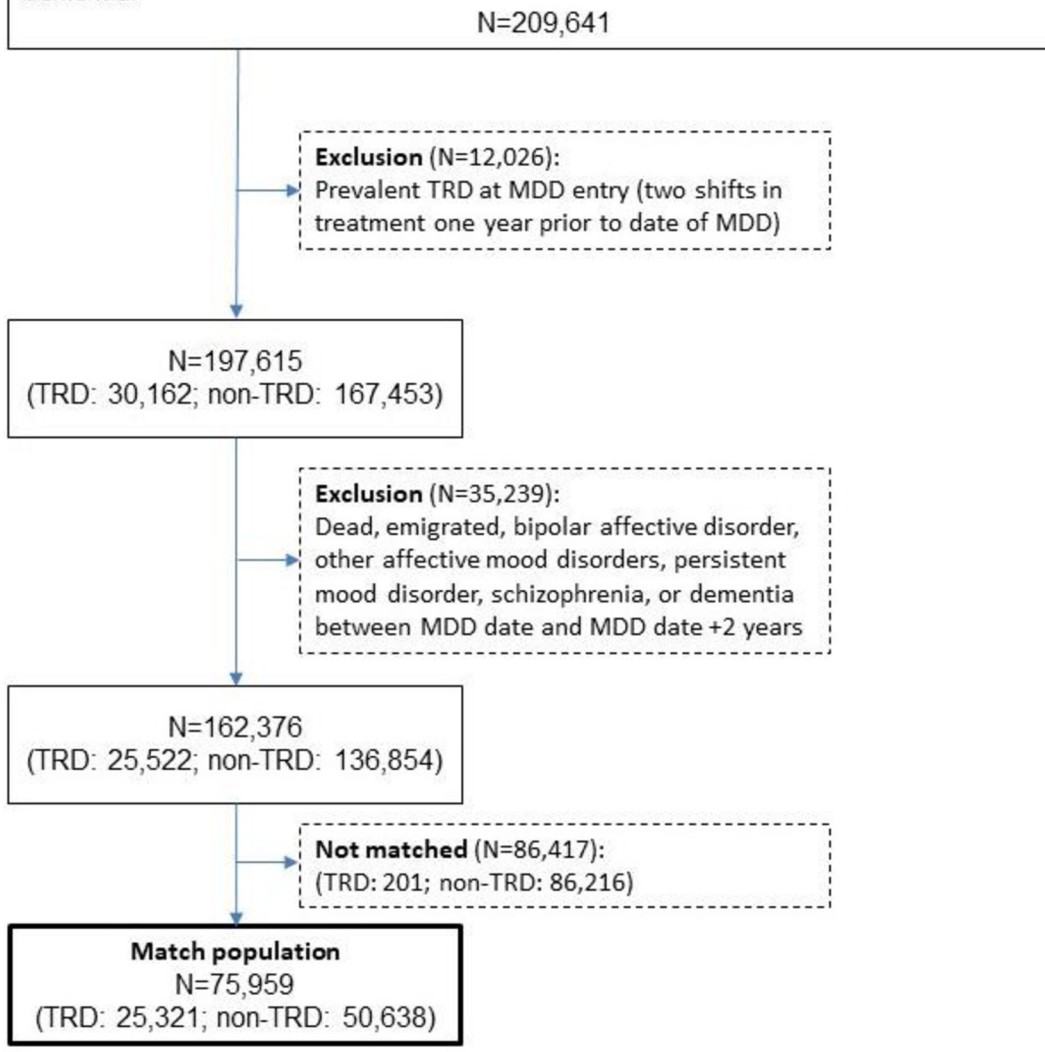

**Fig 1. Flowchart of the study population.**

substance to another [SSRI, SNRI, TCA or MAOI] at ATC level 5) or the initiation of electro-convulsive therapy (ECT). Add-on treatment with another antidepressant drug to the existing therapy was also considered a treatment shift. Information on redeemed antidepressant pre-scriptions from pharmacies was retrieved from the Danish National Prescription Registry [23] (using ATC codes SSRI [N06AB*], SNRI [N06AX*], TCA [N06AA*], MAOI [N06AF*, N06AG*]). Information on ECT was retrieved from the Danish National Patient Register using the Danish Health Classification System codes (SKS: BRTB1 or BRXA1).

## Match population

To construct a matched population, the entire MDD population was followed in national reg-isters one year prior to and one year after MDD diagnosis to categorize patients into TRD and non-TRD populations (S1 Fig in S1 File). For each TRD patient, two patients from the non-TRD MDD population were included as matched references. MDD patients were matched on age (grouped as 18–24, 25–44, 45–64, 65–84 and 85+), sex, year of depression diagnosis (1996–2000, 2001–2005, 2006–2010 and 2011–2015), and depression severity (mild [ICD-10: F32.0, F32.8, F32.9, F33.0, F33.4, F33.8, F33.9], moderate [ICD-10: F32.1, F33.1], or severe [ICD-10: F32.2, F32.3, F33.2, F33.3]). TRD patients were followed for HRU outcomes for a period of one year prior to the date of redeeming the TRD-defining treatment (index date) to one year past the index date. Non-TRD patients were assigned an index date defined as the same number of days since the date of depression diagnosis of the matched TRD patient and had similar follow-up (S1 Fig in S1 File). To ensure complete follow-up for all patients, the cohort was restricted to patients being alive, not having emigrated, and not having developed any of the excluding diagnoses.

## Measures

The Danish national registries capture the entire Danish population, and they include data on HRU which can be linked to demographic and economic data on an individual level.

**Descriptive variables.** Information on sex, age (date of birth) and cohabitation (yes/no) was retrieved from the Central Population Register [24]. Level of education at the time of the depression diagnosis, retrieved from the Attainment Register [25], was investigated and cate-gorized as basic (elementary school 9th grade or less), medium (high school, college, vocational education, short-term higher education), high (medium or long-term higher education) and unknown. Diagnoses with anxiety (ICD-10: F40-F48) and substance abuse disorder (F10-F19) within 5 years prior to index date were retrieved from the Danish National Patient Register.

**Healthcare resource utilization.** Three categories of HRU were retrieved from the Dan-ish National Patient Register and the National Health Service Register. 1) Psychiatric HRU included any hospitalization in a psychiatric hospital department (elective or acute), hospital bed days (elective or acute), emergency department (ED) visits, outpatient hospital visits, home visits (visits by psychiatric health personnel in the private home of the patient), private psychiatrist visits, and private psychologist visits. 2) Somatic HRU included any hospitalization in a somatic hospital department (elective or acute), hospital bed days (elective or acute), ED visits, and outpatient hospital visits. 3) General practitioner (GP) visits or other healthcare spe-cialist visits. Data on home visits, GP visits, private psychiatric and private psychologist visits was available in full only from 2006 onwards, due to inherent changes to the national registries containing these data; home visits were incompletely registered in the National Patient Regis-try prior to 2005, and GP visits, private psychiatric and private psychologist visits were differ-ently registered in the National Health Insurance Registry before 2005, causing a break in data across this time point.

**Healthcare resource utilization costs.** Hospital healthcare expenses (hospitalizations, ED visits, and outpatient visits) were calculated using fixed prices for each type of contact (S1 Table in S1 File); or using Diagnostic Related Group (DRG) codes from the Danish National Patient Register [26] available between the period of 2006 to 2017. Using fixed prices allowed for including data for the entire study period while accounting for inflation. DRG codes were used to collect detailed prices for all contacts. Prices for GP, private psychologist, private psychiatrist and other private primary healthcare specialist visits were retrieved from the National Health Service Register [27]. Medicine expenses (total transaction price, comprising the part covered by public subsidies and that paid by the patient) were retrieved from the Danish National Prescription Registry; the costs were presented as under antidepressant medicine (ATC code N06A), other psychiatric medicine (ATC codes N05A and N05B), and other non-psychiatric related medicine.

## Statistical analyses

Results for HRU are presented as the mean with corresponding confidence interval before and after the index date for TRD and non-TRD MDD patients. Due to the bimodal distribution of the data with a fraction of zero-values (representing non-users), and for the positive part (users), a distribution of counts or costs, differences between HRU for TRD and non-TRD patients after the index date were analyzed with a two-part model. First, the risk difference of any use of the HRU in question was tested using a Poisson regression model with robust error variances [28], and results were given as the increased risk percentage of having at least one instance of HRU for TRD patients versus non-TRD patients ([relative risk -1] × 100). Second, the difference in average utilization was tested restricted to patients having at least one utilization using an ANOVA model adjusted for the matching variables. As some patients had a disproportionately high degree of HRU, the data was log transformed prior to analysis. After back transforming from the log scale, results were given as percentage of change in utilization in the TRD group versus the non-TRD MDD group.

Healthcare expenses in euros (conversion rate from Danish kroner: 7.47) for individual patients were calculated separately for the year prior to and for the year after the index date, and presented with corresponding standard deviations as total costs, and stratified into five categories: psychiatric services, somatic services, GP visits, other private primary healthcare specialist visits, and medicine expenses. Differences in costs between TRD and non-TRD patients after the TRD index date were analyzed with the same models used for analyzing HRU.

**Subgroup analyses.** To study if results were dependent on depression severity, the analyses were stratified for mild, moderate or severe depression based on ICD-10 codes assigned at the first hospital contact (defining entry into the MDD cohort). To study time trends, the analyses were stratified on the year of depression-defining first hospital contact collapsed into 5-year intervals.

All analyses and data management tasks were performed using SAS version 9.4 (SAS Institute Inc., Cary, NC, USA).

## Ethics

According to Danish law, ethical permission and informed consent of the study subjects are not required for anonymized registry-based studies. All data was pseudonymized before access of the data handlers. This study was approved by the Danish Data Protection Agency (ID# CSU-FCFS-2016-012 at Statistics Denmark).

## Results

### Baseline characteristics

The MDD population included 209,641 patients, of which 12,026 were excluded due to prevalent TRD at MDD entry, resulting in 30,162 patients fulfilling the criteria for TRD, and 167,453 non-TRD patients. After matching and conditioning, there were 25,321 TRD patients matched with 50,638 non-TRD patients; 0.8% (201) of eligible TRD cases could not be matched (Fig 1).

The percentage of women was slightly higher among TRD patients (64.3%) than non-TRD patients (61.5%), and a smaller fraction was living alone (45.6% vs. 53.0%). Furthermore, compared with non-TRD patients, more TRD patients were diagnosed with moderate (37.8% vs. 29.7%) and severe depression (21.6% vs. 10.5%). Matching abrogated the differences in sex, age, depression severity and year of depression, although slight differences in the distribution of education level and cohabitation remained. The prevalence of previous hospital contacts with anxiety was slightly higher among patients with TRD, whereas the prevalence of substance abuse was slightly lower (Table 1).

### Healthcare resource utilization

In the year following the index date, the risk of having a psychiatric contact was significantly greater for TRD patients than for the matched non-TRD patients. The higher risk of psychiatric HRU was 138.4% (95% confidence interval: 128.3; 149.0) for hospitalization, 120.6% (113.4; 128.1) for hospital bed days, 134.6% (124.1; 145.7) for home visits, and 131.0% (121.1; 141.4) for private psychiatrist visits; the mean psychiatric hospitalization rate was 0.3 for TRD patients versus 0.1 for non-TRD patients, and the mean for hospital bed days was 8.2 versus 4.0 (Table 2).

Among those who had at least one event of hospitalization, the average number of hospitalizations was 6.1% (3.6; 8.7) higher for TRD patients. However, among those who had at least one hospital bed day, the average number of hospital bed days was 29.0% (24.9; 32.9) lower for TRD, than for non-TRD patients (Table 2).

Compared with non-TRD patients, there was a moderately higher risk of 2.1% (1.5; 2.6) for GP visits in the TRD population, accompanied by a 22.0% (20.3; 23.6) higher rate of GP visits among patients that saw a GP at least once. No estimates for the risk difference of having somatic HRU were larger than 10%. The risks of acute somatic hospitalization and having acute somatic hospital bed days were 5.1% (2.1; 8.1) and 3.3% (0.4; 6.2) higher in TRD patients, respectively. However, among patients with hospital bed day utilization, the average number of somatic hospital bed days was 14.6% (11.7; 17.4) lower for TRD patients than non-TRD patients (Table 2).

### Healthcare resource utilization costs

Compared with non-TRD patients, the total healthcare costs were higher after the TRD index date for TRD patients (Table 3). The total average costs per patient, including medicine expenses in the year following the index date, were €10,543 for TRD patients and €6,927 for non-TRD patients. TRD patients that incurred costs had total expenditures that were 101.9% (97.5; 106.4) higher than non-TRD patients (Table 3). There was an increased risk of TRD patients having any healthcare costs after the index date for psychiatric, somatic, GP and other private healthcare specialist categories; of which the risks of having a psychiatric cost (54.9% [53.3; 56.6] higher risk) and depression medicine cost (32.0% [31.3; 32.6] higher risk) were the highest. Patients incurring a cost saw an increase in expenses for psychiatric (55.2% [50.9;

**Table 1. Background characteristics of the TRD and non-TRD populations before and after matching.**

| Variable | MDD[1] population | | Match population | | p-value[3] |
|---|---|---|---|---|---|
| | TRD[2] | Non-TRD | TRD | Non-TRD | |
| **Sex** | | | | | |
| Women | 19,407 (64.3%) | 103,030 (61.5%) | 16,546 (65.3%) | 33,089 (65.3%) | |
| Men | 10,755 (35.7%) | 64,423 (38.5%) | 8,775 (34.7%) | 17,549 (34.7%) | 0.998 |
| **Age** | | | | | |
| Average age | 49.3 (36.1; 67.0) | 48.2 (32.3; 70.5) | 47.7 (35.4; 63.0) | 47.6 (34.0; 62.9) | |
| 18–24 | 2,523 (8.4%) | 21,349 (12.7%) | 2,167 (8.6%) | 4,334 (8.6%) | |
| 25–44 | 10,078 (33.4%) | 54,519 (32.6%) | 9,081 (35.9%) | 18,162 (35.9%) | |
| 45–64 | 9,396 (31.2%) | 41,350 (24.7%) | 8,292 (32.7%) | 16,583 (32.7%) | |
| 65–84 | 6,970 (23.1%) | 38,135 (22.8%) | 5,054 (20.0%) | 10,106 (20.0%) | |
| 85+ | 1,195 (4.0%) | 12,100 (7.2%) | 727 (2.9%) | 1,453 (2.9%) | 0.989 |
| **Education level at depression diagnosis** | | | | | |
| Basic | 11,267 (37.4%) | 66,174 (39.5%) | 9,373 (37.0%) | 19,115 (37.7%) | |
| Medium | 11,942 (39.6%) | 60,156 (35.9%) | 10,272 (40.6%) | 19,451 (38.4%) | |
| High | 4,572 (15.2%) | 22,851 (13.6%) | 3,960 (15.6%) | 8,085 (16.0%) | |
| Unknown | 2,381 (7.9%) | 18,272 (10.9%) | 1,716 (6.8%) | 3,987 (7.9%) | <0.0001 |
| **Cohabitation** | | | | | |
| No | 13,752 (45.6%) | 88,708 (53.0%) | 11,284 (44.6%) | 24,654 (48.7%) | |
| Yes | 16,277 (54.0%) | 77,465 (46.3%) | 13,930 (55.0%) | 25,583 (50.5%) | |
| Unknown | 133 (0.4%) | 1,280 (0.8%) | 107 (0.4%) | 401 (0.8%) | <0.0001 |
| **Year of depression diagnosis** | | | | | |
| 1996–2000 | 4,681 (15.5%) | 27,806 (16.6%) | 3,860 (15.2%) | 7,720 (15.2%) | |
| 2001–2005 | 7,532 (25.0%) | 39,682 (23.7%) | 6,313 (24.9%) | 12,625 (24.9%) | |
| 2006–2010 | 9,593 (31.8%) | 48,973 (29.2%) | 8,018 (31.7%) | 16,035 (31.7%) | |
| 2011–2015 | 8,356 (27.7%) | 50,992 (30.5%) | 7,130 (28.2%) | 14,258 (28.2%) | 1.00 |
| **Depression severity** | | | | | |
| Mild | 12,225 (40.5%) | 100,160 (59.8%) | 10,034 (39.6%) | 20,068 (39.6%) | |
| Moderate | 11,407 (37.8%) | 49,705 (29.7%) | 9,962 (39.3%) | 19,924 (39.3%) | |
| Severe | 6,530 (21.6%) | 17,588 (10.5%) | 5,325 (21.0%) | 10,646 (21.0%) | 0.987 |
| **Psychiatric-related other comorbidity[4]** | | | | | |
| Anxiety | N/A | N/A | 7,898 (31.2%) | 14,646 (28.9%) | <0.0001 |
| Substance abuse | N/A | N/A | 3,345 (13.2%) | 7,424 (14.7%) | <0.0001 |

[1] MDD: major depressive disorder.

[2] TRD: treatment-resistant depression.

[3] statistical test of TRD vs non-TRD after matching using Chi$^2$.

[4] There was no equivalent index date for the MDD population before matching, hence the prevalences of anxiety and substance abuse disorder were not reported (N/A).

59.6]) and GP services (25.3% [23.1; 27.6]), and for medicine costs (antidepressant medicine: 129.0% [124.2; 133.8]; other psychiatric medicine: 25.2% [21.7; 28.7], other non-psychiatric medicine: 23.3% [20.4; 26.3]) (Table 3). In contrast, somatic costs were 3.8% (6.2; 1.3) lower for TRD patients after index date (Table 3). For the subgroup of patients (from 2006 onward) where DRG costs were available, results were similar (S2 Table in S1 File).

## Subgroup analyses by depression severity and year of diagnosis

For both TRD and non-TRD patients, the risk of psychiatric HRU tended to increase with depression severity (Fig 2), a tendency that was not found for somatic HRU (S2 Fig in S1 File).

**Table 2. Average healthcare resource utilization comparing TRD patients with non-TRD patients in the matched population.**

| Variable | TRD | Non-TRD | TRD vs. non-TRD | | | |
|---|---|---|---|---|---|---|
| | Mean (sd)[1] | Mean (sd)[1] | Change (%) in risk of utilization (95% CI)[2] | p-value | Change (%) in utilization (95% CI)[3] | p-value |
| **Psychiatric contacts** | | | | | | |
| Hospitalization | 0.3 (0.9) | 0.1 (0.5) | 138.4 (128.3; 149.0) | <0.001 | 6.1 (3.6; 8.7) | <0.001 |
| Acute | 0.2 (0.8) | 0.1 (0.5) | 141.4 (130.6; 152.7) | <0.001 | 4.9 (2.4; 7.5) | <0.001 |
| Elective | 0.0 (0.3) | 0.0 (0.1) | 149.0 (122.2; 178.9) | <0.001 | 4.1 (0.2; 8.3) | 0.041 |
| Hospital bed days | 8.2 (27.2) | 4.0 (20.8) | 120.6 (113.4; 128.1) | <0.001 | -29.0 (-32.9; -24.9) | <0.001 |
| Acute | 6.9 (24.3) | 3.2 (17.5) | 120.5 (112.9; 128.4) | <0.001 | -25.3 (-29.5; -20.7) | <0.001 |
| Elective | 1.3 (10.8) | 0.8 (9.9) | 138.7 (117.5; 162.0) | <0.001 | -50.3 (-56.6; -43.0) | <0.001 |
| ED visit | 0.2 (1.2) | 0.1 (0.8) | 85.8 (77.3; 94.7) | <0.001 | 1.6 (-1.4; 4.7) | 0.296 |
| Outpatient visit | 9.3 (16.3) | 4.7 (12.5) | 61.7 (59.5; 64.0) | <0.001 | 34.6 (31.5; 37.7) | <0.001 |
| Home visit | 1.8 (7.2) | 0.6 (3.8) | 134.6 (124.1; 145.7) | <0.001 | 29.9 (23.5; 36.5) | <0.001 |
| Private psychiatrist visit | 0.8 (2.4) | 0.3 (1.4) | 131.0 (121.1; 141.4) | <0.001 | 27.1 (22.5; 31.8) | <0.001 |
| Private psychologist visit | 0.2 (1.1) | 0.2 (1.0) | 32.1 (24.6; 40.1) | <0.001 | -0.4 (-4.9; 4.4) | 0.872 |
| **Somatic contacts** | | | | | | |
| Hospitalization | 0.6 (1.5) | 0.5 (1.5) | 2.8 (0.3; 5.5) | 0.029 | -0.5 (-2.4; 1.4) | 0.580 |
| Acute | 0.4 (1.3) | 0.4 (1.2) | 5.1 (2.1; 8.1) | <0.001 | 0.4 (-1.6; 2.4) | 0.680 |
| Elective | 0.1 (0.5) | 0.1 (0.8) | -4.5 (-9.0; 0.2) | 0.060 | -3.3 (-5.3; -1.2) | 0.003 |
| Hospital bed days | 2.7 (9.9) | 3.4 (12.4) | 0.5 (-1.9; 3.0) | 0.689 | -14.6 (-17.4; -11.7) | <0.001 |
| Acute | 2.0 (7.8) | 2.3 (9.3) | 3.3 (0.4; 6.2) | 0.024 | -11.6 (-14.7; -8.4) | <0.001 |
| Elective | 0.7 (4.6) | 1.1 (6.7) | -8.5 (-12.7; -4.1) | <0.001 | -20.5 (-24.9; -15.9) | <0.001 |
| ED visit | 0.4 (1.0) | 0.4 (1.0) | -2.7 (-5.4; 0.1) | 0.055 | -0.8 (-2.4; 0.8) | 0.334 |
| Outpatient visit | 3.0 (7.0) | 3.2 (8.0) | 2.8 (1.4; 4.3) | <0.001 | -3.8 (-5.8; -1.9) | <0.001 |
| **GP visit** | 9.2 (9.3) | 7.7 (8.2) | 2.1 (1.5; 2.6) | <0.001 | 22.0 (20.3; 23.6) | <0.001 |
| **Other healthcare specialist visit** | 1.9 (3.1) | 1.8 (3.0) | 4.7 (3.5; 6.0) | <0.001 | 2.7 (1.1; 4.4) | 0.001 |

[1] Mean healthcare resource utilization with standard deviation (sd) in the year following the index date.

[2] The percentage change in risk of having any utilization of the particular healthcare service in the year after the index date is given for TRD relative to matched non-TRD patients, presented with 95% confidence interval (CI).

[3] The percentage change in utilization in the year after the index date for TRD patients relative to matched non-TRD patients, among patients who have some healthcare resource utilization.

Considering the relative risk (RR, comparing TRD patients with non-TRD patients) of having HRU, there was a consistent trend of higher RR for patients with mild depression compared with patients with severe depression for all psychiatric healthcare point estimates, except private psychologists (Fig 2). Similarly, the RR of having any psychiatric healthcare expenses or antidepressant medicine related costs was higher for patients with mild versus severe depression (S3 Table in S1 File). However, among patients incurring costs, the relative increase in total HRU costs including medicine expenses did not follow a consistent trend for depression severity (S3 Table in S1 File).

Stratifying on time of diagnosis, the risk of HRU after the index date decreased over the time periods of 1996–2001 and 2006–2015 in both TRD and non-TRD patients for all types of psychiatric HRU, except for outpatient visits. The RR (TRD vs. non-TRD patients) was significant for all periods and tended to increase over time for hospitalizations (overall and acute), hospital bed days (overall and acute), and ED visits (Fig 3). Overall, there was no significant difference between somatic HRU, except for a lower RR of elective hospitalizations (2011–2015, RR: 0.9 [0.8; 1.0]) and elective hospital bed days (2006–2010, RR: 0.9 [0.8; 1.0]; 2011–

**Table 3. Average healthcare resource utilization costs (EUR) comparing TRD patients with non-TRD patients in the matched population.**

| Variable | TRD | Non-TRD | TRD vs. non-TRD | | | |
|---|---|---|---|---|---|---|
| | Mean (sd)[1] | Mean (sd)[1] | Change (%) in risk of cost (95% CI)[2] | p-value | Change (%) in cost (95% CI)[3] | p-value |
| Total, excl. medicine | 9,442 (15,933) | 6,272 (12,924) | 3.6 (3.3; 3.9) | <0.001 | 86.9 (82.4; 91.5) | <0.001 |
| Total, incl. medicine | 10,543 (16,110) | 6,927 (13,091) | 1.2 (1.1; 1.3) | <0.001 | 101.9 (97.5; 106.4) | <0.001 |
| Psychiatric[4] | 6,634 (14,825) | 3,288 (11,238) | 54.9 (53.3; 56.6) | <0.001 | 55.2 (50.9; 59.6) | <0.001 |
| Somatic | 2,581 (5,679) | 2,787 (6,403) | 1.8 (0.6; 3.0) | 0.002 | -3.8 (-6.2; -1.3) | 0.003 |
| GP | 226 (200) | 184 (173) | 1.4 (1.1; 1.7) | <0.001 | 25.3 (23.1; 27.6) | <0.001 |
| Other private healthcare specialist[5] | 105 (266) | 105 (300) | 4.2 (3.1; 5.4) | <0.001 | 2.1 (-0.5; 4.7) | 0.115 |
| Medicine, other | 409 (815) | 349 (760) | 6.2 (5.6; 6.8) | <0.001 | 23.3 (20.4; 26.3) | <0.001 |
| Medicine, other psychiatric | 46 (125) | 27 (105) | 54.3 (51.9; 56.8) | <0.001 | 25.2 (21.7; 28.7) | <0.001 |
| Medicine, antidepressant | 645 (770) | 279 (508) | 32.0 (31.3; 32.6) | <0.001 | 129.0 (124.2; 133.8) | <0.001 |

[1] Mean healthcare resource utilization costs with standard deviation (sd) in the year following the index date.

[2] The percentage change in risk of incurring any cost of the particular type of healthcare expenditure in the year after the index date is given for TRD relative to matched non-TRD patients, presented with 95% confidence interval (CI).

[3] The percentage change in costs in the year after the index date for TRD patients relative to matched non-TRD patients, presented with 95% confidence interval (CI).

[4] Psychiatric cost includes hospital-based psychiatric services, private psychiatrist services, private psychologist services and psychiatric home visits.

[5] Other private healthcare specialist service costs include somatic private therapists (e.g. physiotherapist).

2015, RR: 0.9 [0.8; 0.9]) in the late years comparing TRD to non-TRD patients (S3 Fig in S1 File). The total costs including medicine expenses increased from the first (€10,091) to the second (€11,505) quinquennial for TRD patients, then declined towards the fourth quinquennial (€9,923), but more so in non-TRD patients. Among patients with a healthcare resource cost, the relative difference in costs for TRD patients compared with non-TRD patients increased from 1996–2000 to 2011–2015 for total costs (from 73.3% (62.9%; 84.4%) to 109.3% (101.3%; 117.7%)), particularly for psychiatric costs (from 20.4% (11.1%; 30.5%) to 70.9% (63.3%; 78.9%)) and antidepressant medicine costs (from 46.7% (40.7%; 52.9%) to 173.5% (162.8%; 184.5%)). For somatic hospital costs, they were significantly reduced in the TRD group compared with non-TRD patients only in the early years (1996–2000: -8.4% [-14.9%; -1.5%]; 2001–2005: -5.6% [-10.4%; -0.5%]) (S4 Table in S1 File).

## Discussion

The present Danish nationwide register-based study of patients with a first-time MDD diagnosis comparing patients with TRD to a matched sample of non-TRD patients rendered several findings: 1) TRD patients had a significant 2-fold higher risk of using psychiatric healthcare services during the first year after diagnosis. 2) In contrast, HRU for somatic services was not higher in TRD patients, whereas there was a modest increase in GP visits, which was comprised of both psychiatric and somatic services. 3) Greater HRU also led to greater healthcare costs compared with non-TRD patients, particularly for psychiatric services and antidepressant medicine, and secondly for GP visits and non-antidepressant medicine. 4) The psychiatric HRU increased with depression severity, but the RR of having psychiatric HRU was highest for TRD patients with mild versus severe depression. 5) The RR for having HRU related to psychiatric services among TRD patients increased consistently during the study period. This time trend was particularly evident for acute hospitalizations and ED visits. 6) Finally, the two-part model revealed that whereas the risk of having psychiatry-related hospital bed days was more than 2-fold higher in TRD patients, among patients with hospital bed days, the average number of hospital bed days was lower in TRD patients.

| Variable | TRD[1] | Non-TRD[1] | Relative Risk (95% CI)[2] | | p-value |
|---|---|---|---|---|---|
| **Hospitalization** | | | | | |
| Mild | 13% | 5% | 2.8 (2.6; 3.0) | | <0.001 |
| Moderate | 16% | 7% | 2.3 (2.2; 2.5) | | <0.001 |
| Severe | 22% | 10% | 2.1 (1.9; 2.2) | | <0.001 |
| **Acute** | | | | | |
| Mild | 12% | 4% | 2.9 (2.6; 3.1) | | <0.001 |
| Moderate | 14% | 6% | 2.3 (2.2; 2.5) | | <0.001 |
| Severe | 20% | 9% | 2.1 (2.0; 2.3) | | <0.001 |
| **Elective** | | | | | |
| Mild | 2% | 1% | 2.7 (2.2; 3.4) | | <0.001 |
| Moderate | 3% | 1% | 2.6 (2.2; 3.2) | | <0.001 |
| Severe | 4% | 2% | 2.2 (1.8; 2.6) | | <0.001 |
| **Hospital bed days** | | | | | |
| Mild | 18% | 6% | 3.0 (2.8; 3.2) | | <0.001 |
| Moderate | 22% | 10% | 2.2 (2.1; 2.3) | | <0.001 |
| Severe | 36% | 20% | 1.8 (1.7; 1.9) | | <0.001 |
| **Acute** | | | | | |
| Mild | 16% | 5% | 3.0 (2.8; 3.3) | | <0.001 |
| Moderate | 20% | 9% | 2.2 (2.0; 2.3) | | <0.001 |
| Severe | 33% | 19% | 1.8 (1.7; 1.9) | | <0.001 |
| **Elective** | | | | | |
| Mild | 2% | 1% | 2.8 (2.3; 3.3) | | <0.001 |
| Moderate | 4% | 2% | 2.6 (2.2; 3.0) | | <0.001 |
| Severe | 6% | 3% | 2.0 (1.7; 2.3) | | <0.001 |
| **ED visits** | | | | | |
| Mild | 11% | 5% | 2.1 (2.0; 2.3) | | <0.001 |
| Moderate | 12% | 7% | 1.8 (1.6; 1.9) | | <0.001 |
| Severe | 14% | 8% | 1.7 (1.5; 1.8) | | <0.001 |
| **Outpatient visit** | | | | | |
| Mild | 50% | 25% | 2.0 (2.0; 2.1) | | <0.001 |
| Moderate | 69% | 45% | 1.5 (1.5; 1.6) | | <0.001 |
| Severe | 78% | 55% | 1.4 (1.4; 1.4) | | <0.001 |
| **Home visit** | | | | | |
| Mild | 10% | 3% | 3.1 (2.9; 3.4) | | <0.001 |
| Moderate | 12% | 6% | 2.2 (2.0; 2.3) | | <0.001 |
| Severe | 18% | 9% | 2.0 (1.9; 2.2) | | <0.001 |
| **Private psychiatrist visit** | | | | | |
| Mild | 17% | 6% | 2.7 (2.5; 2.9) | | <0.001 |
| Moderate | 14% | 6% | 2.2 (2.1; 2.4) | | <0.001 |
| Severe | 12% | 6% | 1.8 (1.6; 2.0) | | <0.001 |
| **Private psychologist visit** | | | | | |
| Mild | 6% | 4% | 1.5 (1.4; 1.7) | | <0.001 |
| Moderate | 7% | 6% | 1.2 (1.1; 1.3) | | <0.001 |
| Severe | 6% | 5% | 1.2 (1.1; 1.4) | | 0.001 |

0.5   2   4

RR[3]

**Fig 2. Psychiatric healthcare resource utilization by depression severity.** [1] The fraction of users of the given psychiatric healthcare service for TRD patients and non-TRD patients of the match population in the year following the index date. [2] The relative risk (RR) estimate comparing TRD with non-TRD patients in each stratum is presented with 95% confidence interval (CI) and P-value. [3] Vertical line represents RR = 1.

| Variable | TRD[1] | Non-TRD[1] | Relative Risk (95% CI)[2] | | p-value |
|---|---|---|---|---|---|
| **Hospitalization** | | | | | |
| 1996-2000 | 19% | 9% | 2.0 (1.8; 2.2) | | <0.001 |
| 2001-2005 | 18% | 8% | 2.4 (2.2; 2.6) | | <0.001 |
| 2006-2010 | 15% | 6% | 2.5 (2.3; 2.7) | | <0.001 |
| 2011-2015 | 13% | 5% | 2.7 (2.4; 2.9) | | <0.001 |
| **Acute** | | | | | |
| 1996-2000 | 16% | 8% | 2.0 (1.8; 2.2) | | <0.001 |
| 2001-2005 | 16% | 7% | 2.4 (2.2; 2.6) | | <0.001 |
| 2006-2010 | 14% | 5% | 2.5 (2.3; 2.8) | | <0.001 |
| 2011-2015 | 13% | 5% | 2.7 (2.5; 3.0) | | <0.001 |
| **Elective** | | | | | |
| 1996-2000 | 4% | 2% | 2.3 (1.8; 2.8) | | <0.001 |
| 2001-2005 | 3% | 1% | 2.6 (2.1; 3.2) | | <0.001 |
| 2006-2010 | 2% | 1% | 2.6 (2.1; 3.2) | | <0.001 |
| 2011-2015 | 2% | 1% | 2.6 (2.0; 3.4) | | <0.001 |
| **Hospital bed days** | | | | | |
| 1996-2000 | 27% | 15% | 1.8 (1.6; 1.9) | | <0.001 |
| 2001-2005 | 27% | 12% | 2.2 (2.1; 2.3) | | <0.001 |
| 2006-2010 | 23% | 10% | 2.4 (2.2; 2.5) | | <0.001 |
| 2011-2015 | 19% | 8% | 2.5 (2.3; 2.7) | | <0.001 |
| **Acute** | | | | | |
| 1996-2000 | 23% | 13% | 1.7 (1.6; 1.8) | | <0.001 |
| 2001-2005 | 24% | 11% | 2.2 (2.0; 2.3) | | <0.001 |
| 2006-2010 | 21% | 9% | 2.4 (2.2; 2.5) | | <0.001 |
| 2011-2015 | 18% | 7% | 2.5 (2.3; 2.7) | | <0.001 |
| **Elective** | | | | | |
| 1996-2000 | 6% | 3% | 2.0 (1.6; 2.3) | | <0.001 |
| 2001-2005 | 4% | 2% | 2.6 (2.2; 3.1) | | <0.001 |
| 2006-2010 | 3% | 1% | 2.5 (2.1; 3.0) | | <0.001 |
| 2011-2015 | 2% | 1% | 2.7 (2.1; 3.4) | | <0.001 |
| **ED visit** | | | | | |
| 1996-2000 | 11% | 7% | 1.6 (1.4; 1.8) | | <0.001 |
| 2001-2005 | 12% | 7% | 1.7 (1.6; 1.9) | | <0.001 |
| 2006-2010 | 12% | 6% | 1.9 (1.8; 2.1) | | <0.001 |
| 2011-2015 | 13% | 6% | 2.1 (1.9; 2.3) | | <0.001 |
| **Outpatient visit** | | | | | |
| 1996-2000 | 62% | 39% | 1.6 (1.5; 1.6) | | <0.001 |
| 2001-2005 | 61% | 36% | 1.7 (1.6; 1.7) | | <0.001 |
| 2006-2010 | 60% | 37% | 1.6 (1.6; 1.7) | | <0.001 |
| 2011-2015 | 70% | 45% | 1.6 (1.5; 1.6) | | <0.001 |
| **Home visit** | | | | | |
| 2006-2010 | 22% | 9% | 2.3 (2.2; 2.5) | | <0.001 |
| 2011-2015 | 17% | 7% | 2.3 (2.1; 2.5) | | <0.001 |
| **Private psychiatrist visit** | | | | | |
| 2006-2010 | 22% | 10% | 2.3 (2.2; 2.4) | | <0.001 |
| 2011-2015 | 17% | 8% | 2.2 (2.0; 2.3) | | <0.001 |
| **Private psychologist visit** | | | | | |
| 2006-2010 | 9% | 7% | 1.3 (1.2; 1.4) | | <0.001 |
| 2011-2015 | 12% | 9% | 1.3 (1.2; 1.4) | | <0.001 |

0.5   2   4

RR[3]

**Fig 3. Psychiatric healthcare resource utilization by year of MDD diagnosis.** [1] The fraction of users of the given psychiatric healthcare service for TRD patients and non-TRD patients of the match population in the year following the index date. [2] The relative risk (RR) estimate comparing TRD with non-TRD patients in each stratum is presented with 95% confidence interval (CI) and P-value. [3] Vertical line represents RR = 1.

## Comparison with other studies

Despite fundamental differences in healthcare systems, our findings of higher HRU and HRU-related costs in TRD patients compared with non-TRD patients are in likening with the literature [16, 17] of which the majority of comparable studies were based on medical claims data from commercial insurance registers in the US. Few studies have been conducted in Europe. A recent study [29] analyzing self-reported questionnaire data from internet-based health surveys in five European countries identified 622 TRD patients, who had three times higher odds of overall HRU within the past 6 months than non-TRD patients with depression. The survey did not differentiate between psychiatric and somatic HRU, where we find that the association with TRD was opposite for psychiatric and somatic HRU. The literature is indeed ambiguous in respect to the association between TRD and somatic disease. Two European multicenter studies found no difference in the prevalence of somatic comorbidities compared to non-TRD MDD patients [30–32], whereas, another Danish cohort study found lower risk of TRD for a range of frequent somatic conditions such as cardiovascular disease, infectious disease and (although not significantly) lower respiratory diseases and diabetes in a Danish population study. The authors speculated that some medical conditions may preclude MDD patients from fulfilling the criteria for TRD simply because of contraindications (Gronemann) [15]. In contrast, a recent Danish cohort study, however, found increased odds ratios for cardiovascular, endocrine and neurological conditions in both men and women, and musculoskeletal and hematological conditions in men following the first anti-depressant trial comparing TRD to non-TRD MDD; for none of the reported disease categories were the odds reduced [33]. Notably, the authors of the latter study included prescription medicine to define disease burden, whereas the HRU risk analyses in our study was merely based on health care contacts.

**Depression severity.** While the psychiatric HRU was highest for severe depression for both TRD and non-TRD patients, the largest difference in psychiatric HRU between TRD and non-TRD was observed in patients with mild depression. This was corroborated by Gronemann *et al.* who found that among Danish TRD patients, those with mild depression posed higher relative rates of suicide, compared to non-TRD patients [11]. Mild depression is usually treated in the primary care, and psychiatric hospital treatment is, therefore, less common. However, as the depression-defining hospital contact for some patients might have been due to a somatic primary cause, complicated by depression, one could speculate that the patients with mild depression in the present study constituted a more heterogeneous group than patients with severe depression, with a wider spread of psychiatric treatment needs. Hence, among patients with mild depression, the anti-depressive treatment use defined by the TRD algorithm may have been a stronger discriminator of psychiatric-related treatment needs, more so than among patients diagnosed with severe depression, in whom switch in anti-depressive medicine or ECT (with assumed underlying cause of poor treatment response) was less of a defining factor of psychiatric treatment.

**Hospital bed days.** The present study found a 2-fold increased risk of hospital bed days, but among hospitalized patients, the mean number of bed days was 29% lower, i.e. TRD patients had more frequent psychiatric hospital admissions, but of shorter length than non-TRD patients. Interestingly, this pattern was seen for both acute and elective hospitalizations. Whereas the higher rates of acute hospital contacts among TRD patients may indicate a failure of the healthcare sector to respond to the needs of these patients, the shorter but more frequent and planned psychiatric hospital contacts may, in contrast, reflect an attentive clinical follow-up. Further studies are warranted to elucidate the underlying mechanisms of this pattern of utilization.

**Underlying trends in healthcare in Denmark.** Interestingly, the absolute risk of psychiatric hospitalization or having hospitalization days decreased over the study period in both TRD

and non-TRD patients for all types of services, except for outpatient visits, which increased from the first to the last quinquennial. This is reflective of a general trend in hospital care in many European countries [34]. In Denmark in 2008–2017, the growth in outpatient hospital contacts (somatic and psychiatric) was twice that of inpatient contacts, while the number of hospital bed days decreased by 18% [35]. For psychiatric hospitalization from 2009 to 2016, the frequency of hospitalizations per adult patient increased by 12%, but the average stay declined by 19% [36]. Also reflected in the present study, general psychiatric hospital expenses per patient have declined by 8% between 2010 and 2016 [36], whereas the overall per-population public healthcare spending between 2000 and 2017 increased by 35% [35]. On the whole, public subsidies to private medicine expenses (including medical devices) have increased slightly from €1,900 million in 2008 to €2,000 million in 2017 [35]. The sales in pecuniary terms of all medicinal products from pharmacies to adults in Denmark have increased by 28% from 1999 to 2017, whereas the sales of antidepressant medicine (defined by ATC group, not accounting for individual therapeutic indication) have declined by 66% in the same period (S4 Fig in S1 File) [37]. To that end, we find that medicine expenses (out-of-pocket payment and public subsidy aggregated) declined over the study period for all categories of medicine for both TRD and non-TRD patients.

## Strengths and weaknesses

The nationwide Danish health registries have a very high degree of completeness, allowing a detailed analysis of the HRU by type of health care service, and capturing the entire population in regards of any type of public and most private healthcare utilization including medicine. Some data items were only available from 2006 onward, due to changes in registries before this year; restricting the analysis to the index period 2006–2015, ensuring complete coverage of data, did not change the estimates. As healthcare is tax-financed in Denmark, there is negligible economic or social selection bias in the registers, rendering our study highly generalizable.

As the study only included individuals with hospital contacts for depression, the depression diagnoses were verified by medical specialists, however, most likely at the cost of not including depression cases with less complicated symptoms and disease course. A Danish survey [38] found that 10% of individuals scoring moderate to severe on the Major Depression Inventory (MDI) scale did not report any mental healthcare utilization within 180 days after data collection, whereas 47% did not receive other healthcare than that provided by a GP. Such patients treated for depression exclusively via their GP or a private psychiatric specialist, and never referred to hospital for depression were more likely to benefit from the treatment. Taken together, the hospital contact inclusion criteria would select for relatively more treatment failures and more TRD patients into our cohort, potentially also more patients with a severe grade of depression. As the relative difference between TRD and non-TRD was smaller for severe depression, it might be speculated that our study design resulted in an underestimation of the difference between TRD and non-TRD depressive disorder in the entire population.

However, a potentially partly counteracting design effect may be the fact that the MDD population was constructed based on ICD-10 codes. Compared with the DSM-V (or previously DSM-IV) system, which is a frequently used diagnostic tool to define depression, the ICD-10 based definition is known to be more sensitive to the milder spectra of depression [39].

TRD patients were matched to MDD controls on depression severity, whereas we excluded patients with comorbid bipolar affective disorder, other affective mood disorders, persistent mood disorder, schizophrenia, or dementia to ensure that TRD was not caused by other competing diseases, because this might in turn cause non-response to antidepressants, thereby influencing the healthcare resource utilization and its expenses. In sensitivity analyses,

abstaining from the exclusion for other psychiatric disorders, death or migration incident 2 years after the MDD index date did not substantially affect the risk estimates for HRU comparing TRD vs non-TRD (S5 Table in S1 File). However, the reasons for switching treatment, other than lack of positive response, could also have been side effects, contra-indications, adverse drug-drug interactions or the patient's preferences, which are not accounted for in our analyses.

There is no universal definition of TRD [6], although recently most studies have adopted the definition of failure to respond to two consecutive antidepressant treatments. Herein, failure was defined as a shift in antidepressant chemical substance at ATC level 5. Augmentation strategies and psychotherapy were not included in the TRD definition. Whereas some studies take into consideration a minimum duration of antidepressant treatment between shifts, we did not assess treatment duration or the reason for treatment change. Hence, shifts may have occurred within a time span too short to evaluate clinical response and shifts due to side effects and not treatment failure may have been included in the model of TRD. However, a previous study of TRD in the same cohort [22] reassuringly did not find that the observed associations with disease characteristics changed substantially when excluding treatment courses of less than 4 weeks. As an addition or alternative to TRD definition by medication use, some studies [40] evaluate treatment response more directly by assessing clinical symptoms.

## Implications

TRD compared with non-TRD is associated with higher HRU due to the utilization of psychiatric services. Whereas the HRU and incurred costs increase with increasing depression severity in both TRD and non-TRD patients, the relative difference between TRD and non-TRD was largest for patients with mild depression and those diagnosed in the most recent years. The specific causes of the increased acute hospitalizations and emergency department visits in TRD patients should be monitored in follow-up studies and investigated in more detail.

## Supporting information

**S1 File.**
(DOCX)

## Acknowledgments

The authors would like to thank Diana Kali for the language review and proofreading assistance.

## Author Contributions

**Conceptualization:** Frederikke Hørdam Gronemann, Sarah Alulis, Jesper Riise, Nikolaj Bødker, Merete Osler, Janne Petersen.

**Data curation:** Kristoffer Jarlov Jensen, Frederikke Hørdam Gronemann, Mikkel Zöllner Ankarfeldt, Janne Petersen.

**Formal analysis:** Kristoffer Jarlov Jensen, Mikkel Zöllner Ankarfeldt, Janne Petersen.

**Funding acquisition:** Sarah Alulis, Jesper Riise, Nikolaj Bødker.

**Methodology:** Frederikke Hørdam Gronemann, Mikkel Zöllner Ankarfeldt, Espen Jimenez-Solem, Sarah Alulis, Merete Osler, Janne Petersen.

**Supervision:** Merete Osler, Janne Petersen.

**Validation:** Kristoffer Jarlov Jensen, Mikkel Zöllner Ankarfeldt, Janne Petersen.

**Writing – original draft:** Kristoffer Jarlov Jensen.

**Writing – review & editing:** Kristoffer Jarlov Jensen, Frederikke Hørdam Gronemann, Mikkel Zöllner Ankarfeldt, Espen Jimenez-Solem, Sarah Alulis, Jesper Riise, Nikolaj Bødker, Merete Osler, Janne Petersen.

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
