## [Decision Letter · Decision Letter 0]

14 Jul 2022

PONE-D-22-09950Healthcare resource utilization in patients with treatment-resistant depression – a Danish national registry studyPLOS ONE

Dear Dr. Jensen,

Thank you for submitting your manuscript to PLOS ONE. After careful consideration, we feel that it has merit but does not fully meet PLOS ONE’s publication criteria as it currently stands. Therefore, we invite you to submit a revised version of the manuscript that addresses the points raised during the review process.

We look forward to receiving your revised manuscript.

Kind regards,

Matteo Ruggeri

Academic Editor

PLOS ONE

Journal Requirements:

6. Thank you for stating the following financial disclosure:

“The study was financed by Janssen Cilag A/S.”

7. Thank you for stating the following in the Competing Interests section:

“The study was performed by the Copenhagen Phase IV Unit (Phase4CPH) and was financed by Janssen Cilag A/S, which holds license to antidepressant medicine. Via Phase4CPH, JP, EJS and MZA have performed other studies regarding antidepressants involving funding from Janssen Cilag and Eli Lilly, while KJJ, FHG and MO have performed other studies regarding antidepressants involving funding from Janssen Cilag. All funds were given to their institution. JR, NB and SA are employees of Janssen Cilag. Janssen Cilag A/S and its employees did not have access to the registry data located on the servers of Statistics Denmark and had therefore no influence on the data management and data analysis of the present work. The authors at Copenhagen Phase IV Unit retained editorial control of the manuscript.”

Reviewers' comments:

Reviewer's Responses to Questions

**Comments to the Author**

1. Is the manuscript technically sound, and do the data support the conclusions?

Reviewer #1: Yes

Reviewer #2: Yes

2. Has the statistical analysis been performed appropriately and rigorously? 

Reviewer #1: Yes

Reviewer #2: Yes

3. Have the authors made all data underlying the findings in their manuscript fully available?

Reviewer #1: Yes

Reviewer #2: Yes

4. Is the manuscript presented in an intelligible fashion and written in standard English?

Reviewer #1: Yes

Reviewer #2: Yes

5. Review Comments to the Author

Reviewer #1: The rationale for the lack of some data only from 2006 onwards should be mentioned and the consequences of the lack of data should be explained. The lack of data should be included in the weaknesses of the research

In the section “depression severity”, it is difficult to understand the meaning of mild depression as a more heterogeneous group, the argument on anti-depressive treatment should be more elaborated

In the section “Comparison with other studies”, concurrent somatic comorbidities should be better explained, and the contradictions with the existing study should be further explored, the explanation is not sufficient.

Reviewer #2: Dear, I reviewed "Healthcare resource utilization in patients with treatment-resistant depression – a

Danish national registry study". thank you for your working. it's well written and methodologically corrected

6. PLOS authors have the option to publish the peer review history of their article (what does this mean?). If published, this will include your full peer review and any attached files.

Reviewer #1: **Yes: **Alessandro Signorini

Reviewer #2: No

---

## [Author Response · Author response to Decision Letter 0]

15 Aug 2022

We have added to the ethics section that data was pseudonymized and informed consent was not required according to Danish law.

For detailed responses to reviewer comments, please see Rebuttal letter.

---

## [Editor Report · Decision Letter 1]

13 Sep 2022

Healthcare resource utilization in patients with treatment-resistant depression – a Danish national registry study

PONE-D-22-09950R1

Dear Dr. Jensen,

We’re pleased to inform you that your manuscript has been judged scientifically suitable for publication and will be formally accepted for publication once it meets all outstanding technical requirements.

Kind regards,

Matteo Ruggeri

Academic Editor

PLOS ONE

---

## [Editor Report · Acceptance letter]

19 Sep 2022

PONE-D-22-09950R1 

Healthcare resource utilization in patients with treatment-resistant depression – a Danish national registry study 

Dear Dr. Jensen:

I'm pleased to inform you that your manuscript has been deemed suitable for publication in PLOS ONE. Congratulations! Your manuscript is now with our production department. 

Kind regards, 

on behalf of

Dr. Matteo Ruggeri 

Academic Editor

PLOS ONE